# On the Significance of the Terminal Location of Prion-Forming Regions of Yeast Proteins

**DOI:** 10.3390/ijms26041637

**Published:** 2025-02-14

**Authors:** Arthur A. Galliamov, Valery N. Urakov, Alexander A. Dergalev, Vitaly V. Kushnirov

**Affiliations:** A.N. Bach Institute of Biochemistry, Federal Research Center “Fundamentals of Biotechnology” of the Russian Academy of Sciences, Moscow 119071, Russia; arturens96@gmail.com (A.A.G.); valery.urakov@gmail.com (V.N.U.); alexanderdergalioff@gmail.com (A.A.D.)

**Keywords:** prion, amyloid, yeast prions, prion structure, Sup35, Rnq1, proteinase K

## Abstract

The prion-forming regions (PFRs) of yeast prion proteins are usually located at either the N- or C-terminus of a protein. In the Sup35 prion, the main prion structure contains 71 N-terminal residues. Here, we investigated the importance of the terminal PFR location for prion properties. Two prionogenic sequences of 29 and 30 residues and two random sequences of 23 and 15 residues were added to the Sup35 N-terminus, making the original PFR internal. These proteins were overproduced in yeast with two variants of the Sup35 prion. Mapping of the prion-like structures of these proteins by partial proteinase K digestion showed that in most cases, the extensions acquired an amyloid fold, and, strikingly, the prion structure was no longer present or was substantially altered at its original location. The addition of two to five residues to the Sup35 N-terminus often resulted in prion instability and loss when the respective genes were used to replace chromosomal *SUP35*. The structures of yeast prions Mot3, Swi1, Lsb2, candidate prions Asm4, Nsp1, Cbk1, Cpp1, and prions based on scrambled Sup35 PFRs were mapped. The mapping showed that the N-terminal location of a QN-rich sequence predisposes to, but does not guarantee, the formation of a prion structure by it and that the prion structure located near a terminus does not always include the actual terminus, as in the cases of Sup35 and Rnq1.

## 1. Introduction

In yeast, amyloids can be passed from one generation of cells to the next, in which case they are called prions. There are currently about nine known amyloid-based prions in the yeast *Saccharomyces cerevisiae* [1,2]. In addition, about fifty yeast prions have been described that are probably or certainly not based on amyloid [3,4,5], but for this reason, we will not consider them here. In all amyloid prions, their prion-forming regions (PFRs) represent only a portion, often a small portion, of a protein. PFRs tend to be located at the ends of polypeptides: in five prions (Sup35, Ure2, Swi1, Mot3, Nup100), they are at the N-terminus, in two (Rnq1, Lsb2) at the C-terminus, and in only two cases (Cyc8 and Mod5) they are internal. One of the latter two, Mod5, can be excluded from this comparison because, unlike all the others, it does not have a separate PFR. Additionally, its prion core, as defined by proteinase K (PK) mapping, is located within the Mod5 functional domain [6]. A common feature of yeast PFRs is that they are rich in glutamine and asparagine (QN-rich) and natively unfolded [7,8].

Probably the best-studied yeast prion is [*PSI*+], which refers to the prion form of the yeast translation termination factor eRF3, also known as Sup35 [9]. Sup35 contains the C domain (amino acid residues 254–685) essential for translation termination and a non-essential intrinsically disordered NM region that can drive the prion formation and propagation. Prion structures mainly reside in the QN-rich N region (residues 2–123); however, sometimes they extend up to residue 148 (Figure 1A) [10,11]. The M domain (124–253) includes the sequence important for the Ssa1 chaperone binding (residues 143–164) and thus critical for the [*PSI*+] propagation [12]. There are at least 23 variants of Sup35 prion folding that manifest as variant [*PSI*+] phenotypes, but these prions, at least those that appear most frequently, are usually divided into two main classes, “weak” and “strong” [10,13]. “Weak” [*PSI*+] show lower efficiency of nonsense suppression and higher levels of soluble Sup35 than the “strong” ones. Observation of the [*PSI*+] phenotype is facilitated by the *ade1-14* nonsense mutation. *Ade1-14* cells require adenine and form red colonies due to the accumulation of red intermediate of adenine synthesis. [*PSI*+] reduces the level of functional Sup35 and thus impairs translation termination, causing readthrough of *ade1-14* and partial restoration of adenine synthesis. Such cells become adenine-independent and form white or pink colonies. Sup35 prion structures were mapped by partial proteinase K (PK) digestion and were found in the regions 2–72, 90–20, 124–149, and 155–220. However, the amyloid nature of the latter structure is questionable. Only the N-terminal structure is present in all studied prion variants. Surprisingly, the [*PSI*+] phenotype depends on the variants of the N-terminal structure but does not depend on the presence of other structures [10].

The terminal location of PFRs appears to be important for their prion properties. Adding glutathione transferase to the N-terminus of Sup35 makes its PFR internal and completely blocks the prion properties of Sup35 [14]. C-terminal GFP fusion to Rnq1 also negatively affects its prion potential [15]. On the contrary, placing an internal PFR at a terminus can strongly enhance its prion properties. Sup35 has a second PFR at residues 90–120. If this PFR ends up at the C-terminus due to a nonsense mutation, the frequency of its spontaneous conversion to the prion state increases about six thousand times [16]. A similar situation appears to occur with the Cyc8 prion protein when it acquires a nonsense mutation at the end of its QN-rich region [17].

An interesting feature of terminal PFRs is that their regions closest to a terminus appear to be the most important for prion properties and, in particular, the most sensitive to prion-inhibiting mutations. In Sup35, the prion structures can form within at least the first 150 residues [10,11], but only the first 70 or less of them are essential for prion formation and define the variant-specific prion phenotype, and only the first 30 residues of Sup35 in all studied prion variants are fully protected from PK. The Rnq1 protein has a C-terminal QN-rich region of about 280 residues, but the major PK-resistant prion core includes only the last 40 residues [10,15]. The Swi1 protein has a large disordered QN-rich N-terminal region of 385 residues [18], but only the first 37 of them are required and sufficient for prion formation [19]. The prion-eliminating mutations also tend to be located near the N-terminus in Sup35 and Swi1 [20,21,22]. Finally, glycine insertion or proline substitution after residue four of Sup35 (which is actually residue three considering in vivo removal of the first methionine) can interfere with [*PSI*+] propagation [20]. In accordance with this finding, following the mapping of numerous Sup35 and Rnq1 structures, no evidence of PK activity against the extreme terminal residues in prion structures was observed [10,15]. This may suggest that these residues are situated within the prion structure.

The described observations give rise to the question of how the terminal versus internal location of a sequence affects its prion properties and whether the property observed in Sup35 and Rnq1, the complete protease resistance of terminal residues, is a universal phenomenon. To address these questions, we introduced a series of relatively small extensions to the Sup35 N-terminus, rendering the Sup35 PFR internal. This resulted in significant alterations in the amyloid folding of this PFR. A study of prion structures of “scrambled” Sup35 with randomized PFR (residues 3–114, [23]) and of several yeast prionogenic proteins revealed that, although prion structures tend to form near the termini of QN-rich regions, this may not be regarded as a general rule, and the terminal residues are not always PK-resistant.

## 2. Results

### 2.1. N-Terminal Extensions to Sup35 Unpredictably Alter Prion Phenotype

To study the effects of PFR context on Sup35 prion formation, we fused the prionogenic and random sequences shown in Figure 1B to the Sup35 N-terminus. This was accomplished by replacing the chromosomal *SUP35* gene with its extended versions in the [*PSI*+]-S1 and -W2 strains using the CRISPR/Cas9 approach. While the prion was retained in most cases, the prion phenotype manifested as colony color varied within each extension/prion combination. In some cases, prions were lost due to the Sup35 modification, most frequently in the W2 prion variant with His6 extension (Figure 1C).

**Figure 1 ijms-26-01637-f001:**
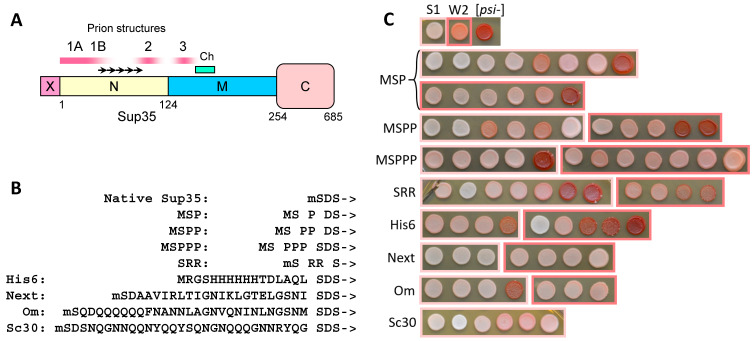
(**A**) Map of the Sup35 protein. X: N-terminal extensions. Prion structures of native Sup35: 1A (residues 2–32), fully PK-resistant; 1B (33–72), partly PK-resistant; 2 (~90–120); 3 (~124–140) [10]. Ch (143–164): Ssa1 chaperone binding site [12]. Oligopeptide repeats are shown by arrows. Amino acid numbering is given at the bottom. (**B**) Sequences of the N-terminal fusions to Sup35 used in this work. The starting methionine is shown by lowercase m when it is removed in vivo. ->: Sup35 sequence starting from the residue five. (**C**) Phenotypes of the cells in which *SUP35* was replaced with the indicated extended *SUP35* variants. The replacement was confirmed by PCR in all presented clones. The spots in light-pink boxes result from the [*PSI*+]-S1 strain, and the spots in dark pink boxes from [*PSI*+]-W2. The [*psi*−] cells lacking prion are red due to the accumulation of red intermediate of adenine synthesis due to *ade1-14* nonsense mutation. In [*PSI*+] cells, this mutation is partially suppressed by the reduced function of Sup35 in translation termination, resulting in pink or white colony color. The “Next” cells did not develop a red color in the absence of prion due to reduced levels of the Next-Sup35 protein (Appendix A).

### 2.2. The Larger N-Terminal Extensions Cause Major Alteration in Amyloid Structure

#### 2.2.1. General Notes

To study the structural alterations caused by the N-terminal extensions, the DNA fragments encoding these extensions were inserted into the pYes2-SUP35NM-GFP multicopy plasmid that was used previously for studying the Sup35 prion structure [10]. This plasmid allows production under the control of the *GAL1* inducible promoter of the Sup35NM protein (residues 1–240) fused to GFP. The obtained plasmids were used for the overproduction of the extended Sup35 proteins, and the structures of these proteins were mapped using partial digestion with proteinase K (PK) and identification of the PK-resistant peptides by mass spectrometry.

It should be noted that despite the availability of prions of extended Sup35 variants, in this work, we mainly studied aggregates of extended Sup35 proteins obtained from cells with native Sup35 prions. The prion nature of these aggregates, i.e., heritability, was not directly proven. We will call aggregates of such kind as amyloids and aggregates as prions when the overproduced protein used for structural mapping is the same as the protein propagating the prion state. However, as we will observe with Om-Sup35, the PK maps did not significantly differ between amyloid and prion preparations. By studying amyloids, we disregard the factors important for prion propagations, such as the ability to be fragmented by the Hsp104 chaperone. Instead, we focus on structural transitions in prion/amyloid. This gave us two advantages. First, it allowed us to show that the structural alterations occur within hours, the timeframe required for galactose-induced expression. Obtaining prions requires two yeast transformations, for chromosomal *SUP35* replacement and for the introduction of overproducing plasmid, and thus takes days. The other advantage relates to the fact that the *SUP35* replacements often result in several different colony colors, which likely represent different prion variants and structures, and this would complicate the interpretation of the data. Finally, the prion maintenance by Next-Sup35 was questionable due to its low level with the *SUP35* promoter, but its production under the *GAL1* promoter was not impaired.

#### 2.2.2. The Om and Sc30 Extensions Dramatically Alter the Original Prion Fold

The Sup35 protein of the yeast *Ogataea methanolica* (*Om*, formerly called *Pichia methanolica*) can acquire the prion state in *S. cerevisiae (Sc)* [24,25], where it forms an N-terminal prion core (Appendix A). The Sup35 PFRs differ substantially in sequence between *Om* and *Sc*, and the prion state is not transferred between the two Sup35 proteins in either direction when they are coexpressed in *Sc* [24]. We added 29 N-terminal residues from *Om* Sup35 before the *Sc* Sup35 sequence as an example of a prionogenic sequence. The aggregates of hybrid Sup35 were obtained in two modes, by overproduction in either the “strong” [*PSI*+] variant S1 or in the same strain with Sup35 replaced for Om-Sup35 and the prion state retained. The obtained maps were largely similar but differed significantly from the original [*PSI*+]-S1 (Figure 2). The added *Om* region acquired full PK resistance, while *Sc* residues 11–46 turned from full to partial PK resistance, and *Sc* residues starting from 46 lost their partial PK resistance. Thus, the N-terminal prion structure apparently shifted towards the N-terminus while retaining its original size of about 70 residues. We presume, as most likely, the following sequence of events. As we mentioned, prion folds are not transferred between *Sc* and *Om* Sup35 due to sequence dissimilarity. So, the prion fold was transferred first to the *Sc* part of Om-Sup35. As Krishnan and Lindquist showed, when two Sup35 molecules are cross-linked in the register at a single point within PFD, they acquire an amyloid fold without noticeable delay [26]. In a similar way, the amyloid fold could be acquired by the *Om* extension. Finally, and most interestingly, structure formation in the *Om* extension changed the pre-existing structure in the *Sc* region.

Here, an important question is what kind of structure has the partially PK-resistant regions. In the original [*PSI*+], this Sup35 region, spanning residues 33–72 (Region 1B), can be separated from Region 1A (2–32) by PK cuts at residues 33, 35, 38, or 41 and then it rapidly degrades, such that we observe little or no fragments belonging solely to the Region 1B. This allows us to suggest that the true amyloid cross-beta fold is formed only in Region 1A, while Region 1B is protected from PK only through its association with 1A. In the Om-Sup35 prion structure, we observed moderate quantities of peptides belonging solely to the partially PK-resistant region 11–46. These peptides appear unlikely to originate from PK cleavage of the main structure since they end near residue 32, much earlier than the main structure. If our logic is correct, it implies that the N-terminal structure in Om-Sup35 removes the pre-existing prion structure of wild-type Sup35 at Sc residues 11–32 and likely alters the structure at residues 2–10.

Similar observations were made when the first 30 residues of Sc Sup35 were repeated before the Sup35 sequence (Figure 2). This protein, Sc30-Sup35, was overproduced in the strong S1 and weak W2 [*PSI*+] variants. In both cases, again, the protease resistance profile shifted to the new N-terminus by the length of the added sequences. The above scenario for Om-Sup35 is less certain in this case since it is not known whether the original Sup35 prion fold was copied to the first or the second Sc30 repeat. However, the outcome was similar: the prionogenic N-terminal extension destroyed the prion structure in the original Sup35 location.

Thus, we observed that the terminally located prionogenic regions dominate over internal prionogenic regions in defining the prion structure and can interfere with structure formation by the latter.

The PK resistance graphs presented in this work also show distal structures located beyond the N-terminal structure. These structures varied substantially between the experiments, but we do not know their significance. As observed earlier, the [*PSI*+] phenotype mainly, if not solely, depends on the N-terminal core [10]. For these reasons, we do not consider distal structures here and show them in grey.

### 2.3. N-Terminal Random Poorly Amyloidogenic Extensions Can Acquire Amyloid Fold

The His6 and Next extensions were intended as non-amyloidogenic. Since we did not have strong considerations for choosing one or another such sequence, we used some DNAs from our collection. The His6 extension (MRGSHHHHHHTDLATM) was taken from the pQE10-SUP35NM plasmid. To obtain the Next N-terminal EXTension, we allowed translation of the pYes2 plasmid sequence upstream of the Sup35 ORF in pYes2-SUP35NM-GFP by adding DNA encoding three first residues of *SUP35*, MSD and replacing the first methionine of Sc Sup35 for isoleucine to exclude Sc Sup35 synthesis, which otherwise occurred.

The Sup35 protein with the His6 extension was overproduced in the [*PSI*+]-S1 strain, where *SUP35* was replaced for its His6-extended version and analyzed. The His6 extension did not acquire an amyloid fold, and, likely in relation to this, the original Core1 structure was not altered (Figure 3). However, we have noted that the preparations contained nearly equal amounts of Sup35NM-GFP with and without extension. The latter apparently resulted from translation initiation at the Sc starting methionine since the related peptides were easily distinguished by the N-terminal acetylation. The presence of significant levels of non-extended Sup35 could favor the propagation of the original prion fold and thus suppress structure formation in the His6 extension.

To avoid this, we replaced the first methionine of Sc Sup35 with leucine, resulting in His61 extension: MRGSHHHHHHTDLAQL. This protein was produced in the original [*PSI*+]-S1 strain and formed aggregates, whose PK digestion revealed a complex picture with a major group of peptides at residues 1–71 and three minor groups at residues approximately −15 to −1, 16–32, and 38–71 (Figure 3). In general, the minor structures could either result from the digestion of the main structure or represent independent structures existing in minor subpopulations of cells. For the PK-resistant His61 peptides (−15 to −1) and the (38–71) peptides, the most likely origin is separation from a larger structure. However, it is difficult to conclude whether this was a single large structure (−15 to 71) with inner PK-sensitive sites or two independent structures like, for example, −15 to 32 and 1–71. Thus, the His61 extension caused significant alterations in the Sup35 prion structure and, in particular, the appearance of inner PK-sensitive sites.

In the [*PSI*+]-W2 variant, His61-Sup35 formed two structures. The major structure included the His61 extension, while the minor structure excluded it. It is unclear whether the minor structure is of independent origin or results from the digestion of the main structure. The C-termini of the PK-resistant peptides ended somewhat earlier than in the prototype W2 structure, but the difference is insufficiently large to conclude with confidence that the structure in the *Sc* Sup35 1–71 region was significantly altered.

The Next-Sup35NMG protein was overproduced in the [*PSI*+]-S1 and W2 strains and formed aggregates. In both cases, the Next extension acquired full protease resistance, and the whole structure shifted towards the new N-terminus, similar to the cases of Om-Sup35 and N30-Sup35. Thus, one can assume the same sequence of events as with Om-Sup35, but now the structure was formed by random poorly amyloidogenic (according to PLAAC and FoldIndex algorithms, Figure 3) peptide. A significant amount of “minor” structure(s) was observed in the Next-S1 preparation. As we discussed for His61-S1, these structures could be either independent or result from the digestion of the main structure. An interesting detail is that in all structures described above, there were structures starting near Sc residue 2, the same as in original [*PSI*+]-S1 and W2 structures. However, in the next minor structures, there are groups of peptides starting strictly from positions −8 and −3, which suggests at least a local difference from the original [*PSI*+]-S1 and W2 structures. Other minor peptides of Next-S1 started from positions 17 and 39, which is similar to peptides found in His61-S1 preparation.

Thus, the observations with the Next and His61 extensions show that even poorly amyloidogenic sequences could acquire an amyloid fold when located at a protein terminus. The proximity to an old Sc Sup35 amyloid core apparently helps to establish the amyloid fold in these poorly amyloidogenic extensions. Nevertheless, it remains uncertain whether the original core facilitates the propagation of extended proteins, given that the prion structure in the Sc region is significantly diminished, particularly in the Next-Sup35 structures.

### 2.4. Small Extensions Can Alter Prion Structure

The Sup35’s N-terminus is highly sensitive to mutations, and glycine insertion or proline substitution after residue four of Sup35 can interfere with [*PSI*+] propagation [20]. Together with observations that the N-terminal Sup35 residues are fully protected from PK, this could indicate that the N-terminal residues are located inside the prion structure. Then, even small additions to the N-terminus could distort the Sup35 prion structure. To test this, DNA fragments encoding N-terminal extensions of one to three prolines and a change of Asp-3 to a double-arginine were added to plasmid-borne *SUP35NM-GFP*.

Plasmid-encoded extended Sup35 variants were overproduced in the S1 and W2 [*PSI*+] strains, and their PK resistance maps were established. These maps did not significantly differ from the Sup35 map in the same strains, with the exception of mSRR extension (Figure 4 and Appendix A). In both [*PSI*+] strains, mSRR significantly altered the Core 1 structure. Firstly, the PK-resistant regions became smaller. Secondly, point sensitivity to PK appeared after residue 15, which was especially pronounced in the W2 [*PSI*+] (Figure 4).

Some observations were made with proline extensions. The first methionine is usually removed in yeast, when followed by serine, as in Sup35 [10]. However, this did not occur with our proline extensions, probably due to the proline in the third position. The retained methionine was only partially PK-resistant in MSP- and MSPP-Sup35 and PK-sensitive in MSPPP. This detail, despite being small, distinguishes the proline extensions from native Sup35 and the larger extensions described above.

In previous studies, we noted that PK is unable to cut before and after proline. With the MSPPP extension, we saw for the first time that this can occur. This indicates that the cleaved sequence is maximally well-accessible to PK. The N-terminal residues of MSPPP-Sup35 were much better protected from PK when this protein polymerized in the presence of [*PSI*+], compared to its amyloid formed de novo in the presence of [*PIN*+] (Table 1). This supports our earlier suggestion that the Sup35 N-terminus is located inside the prion structure [10]. Then, the structure seeded by the Sup35 [*PSI*+] folds would try to accommodate the MSPPP extension within the prion structure where there is insufficient space for it. The amyloid fold formed de novo with the help of [*PIN*+] would allow MSPPP to be located outside of the prion core, thus resolving the conflict. (See Discussion for more details).

### 2.5. Terminal Location Is Insufficient for Prion Structure Formation by QN-Rich Sequences Despite Increasing Its Probability

To investigate whether or to what extent prion structures tend to appear at a protein terminus, we obtained prions of Sup35 proteins with scrambled N domains. These proteins, in which Sup35 residues 3 to 114 are randomized, were originally used by Ross et al. to show that prionogenicity does not depend on the exact amino acid sequence, provided that the amino acid composition is preserved [27]. The strains kindly provided by Prof. Ross contain *SUP35* chromosomal disruption and scrambled *SUP35* variants on centromeric plasmids. We amplified the scrambled *SUP35* regions from *SUP35-24*, *-25*, *-26*, and *-27* gene variants (Ref) and used them to replace the respective regions of the pYes2-SUP35NM-GFP multicopy plasmid. The obtained plasmids were introduced to yeast strains with analogous scrambled *SUP35* variants. Overproduction of scrambled Sup35NM-GFP was used first to select prions and then to obtain prion aggregates of these proteins in individual clones carrying prion.

Prions were obtained for all scrambled Sup35 variants, except for Sup35-25, and they showed various levels of translational suppression expressed as colony color grades (Appendix A). The prion aggregates were purified, and their structure was mapped by PK (Figure 5). In contrast to wild-type Sup35, the formation of the N-terminal prion structure was not a common property of these prions. For Sup35-24, such structure was observed in just one of the four studied prion isolates. It was associated with a prion with the weakest suppressor phenotype. Both of the Sup35-26 prions showed an N-terminal structure. However, in the Sup35-26-7 prion, there were clearly two overlapping structures that existed in parallel. The resistant peptides representing these structures differ by both their start and end points; hence, they seem unlikely to originate from a single structure. Instead, they are more likely to originate from two structural variants existing in parallel in different cellular subpopulations. Sup35-27 prions all had an N-terminal prion structure ending at residues 64 or 66. However, these structures were of two types: with fully or partly protected N-terminus. This is an important distinction since all studied wild-type Sup35 prion variants have fully protected N-terminus, and the same applies to the Rnq1 C-terminus [10,15]. Finally, we overproduced Sup35-25 in the respective strain, and since the studied strains were [*RNQ*+], Sup35-25 formed amyloid with an N-terminal partly protected structure.

Overall, all scrambled Sup35 variants were able to form N-terminal prion structures; however, for Sup35-24, it was not preferable. One of the three observed Sup35-26 structures was not N-terminal. The reluctance of Sup35-24 to form an N-terminal structure could be related to the presence of three proline residues near the terminus (at positions 9, 12, and 24). Proline is poorly compatible with the beta structure, and single proline substitutions within Sup35 prion cores either interfere with prion propagation or preclude the formation of Sup35 amyloid structures in vitro [11,20].

Observations with scrambled Sup35 prions indicate that the terminal location of a sequence significantly increases its probability to form a prion/amyloid structure. However, it does not guarantee the formation of such structure, even with appropriate amino acid composition.

Mapping of the Sup35-24 prions also produced one unexpected observation. In the Sup35-24-1 prion, the major structure was located at residues 111–135, while other detected peptides were minor. The structure coordinates are approximate since some resistant peptides end slightly outside this region and some slightly within. In this region, only the first four residues, YQQP, belong to the scrambled sequence. Furthermore, since proline (P) is known to break beta structures, it is likely that the other three residues are also not included in the amyloid structure but are protected by the nearby structure. Thus, for one of the scrambled Sup35 proteins, described by Ross et al. [27], its prion structure is located mostly outside of the scrambled sequence. This observation does not affect the results of Ross et al., as other tested scrambled Sup35 proteins formed prion structures within their scrambled regions.

### 2.6. Amyloid Structures of Prion and Candidate Prion Proteins

To find out how prionogenic regions of yeast proteins distribute between terminal and inner regions, we obtained in the [*PIN*+] strain aggregates of the PFR-GFP fusions of the following prion proteins: Mot3, Swi1, Lsb2, and prion candidates according to Alberti et al. [18]: Asm4, Nsp1, Cbk1, Cpp1(Ybr016). All the candidates had an N-terminal QN-rich region. We expected that structures obtained in such a way would be largely similar to prion structures of these proteins because this is true for (1) Rnq1 aggregates obtained in the presence of [*PSI*+], (2) Sup35 aggregates obtained in the presence of [*PIN*+] (Appendix A), and (3) Mot3 aggregates (Figure 6). It is reasonable to assume that amyloids obtained de novo represent a mix of possible structures of a given protein.

The proteins studied in this chapter have different cellular functions. Mot3 is an environmentally responsive transcription factor that modulates a variety of processes, including mating, carbon metabolism, stress response, maintenance of cell wall composition, and pheromone signaling [28]. Swi1 is a subunit of the SWI/SNF complex that regulates transcription by remodeling chromatin [29]. Lsb2 (Pin3) is a negative regulator of actin nucleation-promoting factor activity, a short-lived protein whose levels increase in response to thermal stress [30,31]. Asm4, also known as Nup59, and Nsp1 are FG-nucleoporin components of the central core of the nuclear pore complex [32,33]. Nsp1 forms phase-separated liquid condensates within nuclear pores, and its ability to form amyloid likely relates to this property. Cbk1 is a serine/threonine protein kinase involved in the regulation of cell division [34]. Cbk1 homologs are involved in aging in animals [35]. Cpp1 is a plasma membrane C-terminally anchored protein with unknown biological roles localized on the bud membrane and the mating projection membrane [36].

In the first two PK-mapped prions, Sup35 and Rnq1 [10,15], the prion structure is not just terminally located, but their terminal residues are fully protected from PK. This means that the peptides lacking one, two, or several terminal residues were not found among numerous variants and isolates of the Sup35 prion and two variants of the Rnq1 prion. Therefore, it was of interest to establish whether the fully protected terminal cores are typical of other yeast prions.

Contrary to our expectations, in just four proteins, Asm4, Swi1 Lsb2, and Nsp1, the N-terminal residues were fully protected, with minor reservations (Figure 6 and Figure 7). Thus, the PK-protected terminus is not a general property of prionogenic sequences.

The Mot3 structures in prion and amyloid mode proved to be similar. In both, the N-terminus was only partly protected, which suggests that the terminus is likely not involved in cross-beta structure.

In the Swi1 structure, there were minor amounts of peptides starting from residue 7, which could reflect a weak PK cutting site. The PK-protected region of Swi1 ends at residues 68–73 and thus resembles Sup35 in strong [*PSI*+] by size and shape. For comparison, it was shown earlier that the first 37 residues of Swi1 are sufficient to confer prion properties [19]. This again resembles Sup35: its N-terminal prion core is about 70 residues, but the Sup35(1–40)-GFP fusion can co-aggregate with Sup35 in the majority of [*PSI*+] variants [13]. It is worth noting that the value of 37 Swi1 residues was obtained with high overproduction of Swi1(1–37), using TEF1 promoter instead of SWI1 (the difference in expression levels is more than 100-fold according to https://www.yeastgenome.org/ (accessed on 29 November 2024)).

The Lsb2 case is of particular interest since its C-terminus is not QN-rich, and according to computer predictions, it is neither prionogenic nor prone to unfolding (Figure 6). Nevertheless, it formed a structure with full protection of the C-terminus despite with weak point sensitivity to PK located 15, 17, and 25 residues from the C-terminus. Curiously, this resembles the Next-Sup35 case when a poorly amyloidogenic sequence formed a protected terminal structure. The full PK protection and thus structuring of the Lsb2 C-terminus agrees with earlier findings that the third residue from the C-terminus, asparagine, is critical for prion ability and its change for serine, as in the homologous Lsb1 protein, abolishes prion formation [31].

N-terminal cores of the Asm4 and Nsp1 nucleoporins are similar in that both are significantly smaller compared to that of Sup35 and that minor peptides with uncertain relation to the main structure were observed.

In Cbk1, the amyloid core included residues of approximately 5 to 53 (Figure 7). We tested whether Cbk1 would polymerize without this core in view of the fact that the region with the highest QN content in Cbk1 is not its N-terminus but the residues 200–250. We deleted the EcoNI fragment encoding Cbk1 residues 15–50. The truncated Cbk1-GFP protein almost did not aggregate when overproduced in the [*PIN*+] strain, and we were unable to obtain any structure.

The amyloid structure of Cpp1 was actually not N-terminal, being located between residues 49 and 98. The lack of N-terminal structure could be related in part to the presence of several prolines and insufficiently high QN content in the N-terminal region.

As we noted previously in the Rnq1 study [15], the actual location of amyloid structures in most cases correlates poorly with predictions of computer algorithms. The PLAAC and FoldIndex algorithms showed moderately good correlation, and we added their predictions to Figure 6 and Figure 7. We also compared the PK resistance maps of the (candidate) prion proteins with predictions of the ArchCandy algorithm (Appendix A). The terminal amyloid structures were generally predicted correctly, but the possibility of the internal structures was clearly exaggerated, which highlights the need to account for the terminal and internal location of structures in such algorithms.

## 3. Discussion

Earlier biochemical and genetic data related to Sup35, Rnq1, Swi1, and Lsb2 prions suggest that the terminal location of PFRs of yeast proteins is highly important for their prion potential. In this work, we investigated to what extent this suggestion is correct in general.

The fully protected prion structure at the terminus of a protein is a universal and strictly observed property among the Sup35 and Rnq1 prion variants. However, among the seven tested proteins, only Swi1, Lsb2, Asm4, and Nsp1 reproduced this property with minor reservations. Also, the newly established N-terminal amyloid structures, except for Swi1, proved to be smaller than that of Sup35 and/or include internal sites sensitive to PK. This poses the question of whether larger prion cores reflect more important cellular roles of the [*PSI*+], [*PIN*+], and [*SWI*+] prions.

Four proteins, Sup35 [10], Rnq1 [15], Swi1 [19], and Cbk1 (this work), with large potentially prionogenic regions, were tested for the ability to form prion/amyloid structure when the region of the major terminal structure is removed. In three cases (Sup35, Swi1, Cbk1), the prion/amyloid was unable to form ([10,19] and this work). However, analogous deleted variants of Rnq1 were able to propagate the prion state despite a substantial reduction in Rnq1 levels accompanying these deletions [15].

The study of the prion structures of scrambled Sup35 proteins was moderately in favor of the importance of the terminal location: while prions with N-terminal structures occurred more frequently, prions without N-terminal structures were also observed.

Among the small alterations in the Sup35 N-terminus, the most notable effects were caused by the change of aspartate-3 for two arginines: the size of the PK-resistant N-terminal core reduced, and two internal points sensitive to PK appeared. The aspartate-3 neutralizes the positive charge of the N-terminus; hence, the substitution for arginines made the terminus 2+ charged, which could disrupt the local structure. It is not clear whether this actually occurred since we observed that the PK resistance of the N-terminus was not reduced. An explanation of why this substitution had a significant effect on the Sup35 prion structure, in contrast to the larger MSPPP extension, may be given by the hypothesis presented in the next paragraph.

A study of the relatively large (15–30 residues) extensions to the Sup35 N-terminus produced two unexpected observations. Firstly, even poorly amyloidogenic sequences can acquire PK-protected and likely amyloid structures in such a location. Secondly, such a structure formation in the extension can substantially alter or even destroy the original prion structure in the *Sc* region of the extended Sup35 protein. Thus, terminally located sequences, even random ones, can “steal” the structure from the established Sup35 PFR, the structure that was used to seed the prion-like polymerization. This shows that the terminal location allows a sequence to dominate in defining the amyloid structure over neighboring internal sequences. A possible explanation for this is that the conversion of monomers into amyloid tends to start near their terminus. This assumption fits well with the significant effects of the mSRR extension on the amyloid structure and the dependence of the Lsb2 prion properties on the third residue (N or S) from the Lsb2 C-terminus. The properties of terminal residues that increase their chance of being the starting point of amyloid conversion could be their better accessibility and freedom of movement compared to internal residues. These properties can also explain why nonsense mutations that place an internal Sup35 PFR (residues 90–120) at the C-terminus increase the frequency of de novo prion formation by this protein several thousands of times [16].

Sup35 prions and amyloids, like most other amyloids, have a parallel-in-register structure [37]. In such a structure, each amino acid residue of one protomer is aligned to the same residue of the next protomer. Every protomer occupies a single strand-thick two-dimensional layer along the fibril axis. The “serpentine” model for Sup35 prion [38] (Figure 8A) was the first and possibly the only attempt to predict this folding. However, the serpentine fold is likely to be tolerant to small alterations near the N-terminus and thus would not be able to explain the prion-curing effects of proline or glycine introduction after three N-terminal residues of Sup35 [20]. To explain these effects, it is possible to propose a roll-like model where the N-terminus is located in the center of the structure, and there is insufficient space to accommodate any additional residues (Figure 8B). However, such a structure would not tolerate the 5-residue MSPPP extension, which had only a modest effect on S1 and W2 prions (Appendix A). To reconcile the available data, we propose a structure where the N-terminus is located inside of the prion structure but not deeply (Figure 8C). Then, small extensions like MSPPP can be accommodated without significantly distorting the whole structure, and the pre-existing Sup35 prion structure would define lower exposure of the extension than the structure formed de novo (Figure 8D,E; Table 1).

The terminal location of amyloid-forming regions (AFRs) is important, not just in yeast. It is widely known that a good proportion of human amyloids are formed by peptides resulting from proteolysis of full-sized proteins [39], and a good example is the Alzheimer’s peptide a-beta. Apparently, such peptides cannot form amyloid inside of a protein molecule but acquire such ability when their AFR ends up near the N- or C-end of a peptide because of proteolysis.

The pair of amyloid-forming Orb2 proteins of *Drosophila* provide a bright example of how different amyloidogenicity of the terminal and internal amyloid-forming regions is used by nature [40,41]. Orb2A and Orb2B are synthesized from the same pre-mRNA through alternative splicing. They are identical in their amyloid-forming and functional domains but differ in the N-terminal sequence (Figure 9). The Orb2A/B amyloid binds to a subset of mRNAs lacking polyA tail and initiates elongation of such tails, thus activating these mRNAs for translation and changing properties of a particular synapse. Orb2B is constitutively expressed, and its transition to amyloid is initiated by short-term expression of the Orb2A protein. The eight-residue N-terminal sequence of Orb2A, together with a nearby Q-rich region, allows this protein to rapidly form amyloid. A large non-amyloidogenic N-terminal region of Orb2B reduces its amyloidogenic potential and allows this protein to exist stably in the monomeric form. However, it does not interfere with the formation of amyloid when seeded with Orb2A. This allows the amyloid transition and memorization to be controllable events.

## 4. Materials and Methods

### 4.1. Yeast Strains and Media

Yeast strain 74-D694 [*PIN*+] (MATa *ade1-14 ura3-52 leu2-3,112 his3-Δ200 trp1-289*) and its descendants harboring “strong” [*PSI*+]-S1 and “weak” [*PSI*+]-W2 were used. The strains YER289, YER290, YER292 and YER293 with a common genotype MATα *kar1-1 SUQ5 ade2-1 his3 leu2 trp1 ura3 sup35::KanMx* carry centromeric *LEU2* plasmids with scrambled *SUP35* gene variants *SUP35-24*, *SUP35-25*, *SUP35-26* and *SUP35-27*, respectively. Synthetic complete (SC) media contained 6.7 g/L yeast nitrogen base, 20 g/L glucose or galactose, and required amino acids. For colony color development, SC media contained a reduced amount of adenine (7 mg/L, or 1/3 of standard). Adenine-limiting rich medium (YPDred) contained 5 g/L yeast extract, 20 g/L peptone, 20 g/L glucose, and 20 g/L agar.

### 4.2. Plasmids

The multicopy pYes2 plasmid (Thermo Fisher Scientific, Waltham, MA, USA) was used for the overproduction of proteins driven by the *GAL1* promoter for the subsequent PK structural mapping. The pYes2-Sup35NM-GFP plasmid that codes for the Sup35 first 240 residues was taken as a starting point. The PvuII-BalI(MscI) fragment coding for Sup35 residues 1–155 was removed and replaced with PCR fragment encoding either extended or scrambled Sup35 up to residue 155. These fragments included 20 nucleotides upstream of the PvuII site and 20 nucleotides downstream of BalI for ligase-less joining to the pYes2-Sup35NM-GFP fragment using Vazyme (Nanjing, China) ClonExpress II one-step cloning kit. The scrambled *SUP35* regions were amplified from the YER2** strains by PCR. The genes for yeast prions and prion candidates were amplified from the yeast DNA.

### 4.3. Obtaining of Prions and Genetic Procedures

The 74-D694 cells carrying [*PSI*+]-S1 and -W2 were transformed with CRISPR/Cas9 plasmid pWS171 with the target sequence immediately preceding the Sup35 ORF. To obtain the prion form of scrambled Sup35 proteins, the strains YER289, YER290, YER292, and YER293 were transformed with derivatives of the pYes2-Sup35NM-GFP plasmid encoding the same scrambled Sup35 version as a particular strain. The transformed cells were grown for 36 h in SC- Ura galactose medium and then sown onto an *SC -Ade -Ura* agar plate. The presence of prions in colonies growing on this medium was confirmed by growth in the presence of 3 mM Guanidine HCl.

The Mot3 prion was obtained according to [42]. The *Δrnq1* derivative of the 74-D694 strain was transformed with multicopy *LEU2* plasmid pRS315-SUP35 and pYes2-Mot3-GFP. The transformants were grown for 80 h in *SC- Ura* galactose medium and then sown onto an *SC -Ade -Ura* –Leu agar plate, thus selecting for Sup35 prion or amyloid. The Ade+ colonies from this plate were streaked to single cells on the *SC -Ura* low Ade plate, allowing loss of the *SUP35* plasmid. During this, some cells that carried unstable [*PSI*+] or Sup35 amyloid regained red color and became Ade- and [*psi*−]. The presence of the Mot3 prion was verified by the Mot3-GFP aggregation pattern typical of a prion (Appendix A). Fluorescent images of yeast cells were obtained using an Axioscop 40 FL microscope (Zeiss, Oberkochen, Germany).

### 4.4. Western Blotting

Lysates from logarithmic yeast cultures were separated by SDS-PAGE and subjected to Western blotting with anti-Sup35NM antibody. Blots were decorated with rabbit polyclonal anti-Sup35NM antibody and goat anti-rabbit peroxidase-conjugated secondary antibody (Pierce #31460, Thermo Fisher Scientific, Waltham, MA, USA) and developed with the Thermo Fisher Scientific Supersignal West Dura kit. Excel 2016 was used for statistical calculations of the confidence interval and *p*-value for Appendix A. The confidence interval was obtained using the CONFIDENCE.T function (α = 0.05). The *p*-value was obtained using the T.TEST function.

### 4.5. PK Digestion, Mass Spectrometry and Analysis

Prionogenic proteins of interest were overproduced overnight in 400 mL of galactose SC medium in the presence of either [*PSI*+] or [*PIN*+] prions and acquired aggregated form. The aggregates were isolated and analyzed as described [10]. Briefly, 4 µg of purified amyloid was digested by PK (12 µg/mL) in 20 µL for 1 h at room temperature, PK was inactivated by PMSF, and the reactions were precipitated by 16 µL of acetone, washed with acetone, dissolved in 10 µL water, boiled for 3 min at 110 °C; and analyzed by MALDI-TOF/TOF mass spectrometer UltrafleXtreme (Bruker, Billerica, MA, USA). Peptides were identified by MS-MS and/or as groups of related peaks.

To graphically represent the PK-resistant structures for every residue of Sup35NM, we calculated the PK resistance index using Microsoft Excel as a sum of mass spectral peak areas of peptides that include this residue. The resulting graphs were normalized against their maximum value for each preparation.

## 5. Conclusions

Observations of this work show that the terminal location of a sequence strongly increases its propensity to form amyloid, while the internal location has the opposite effect. This effect is used by Orb2A/B proteins of *Drosophila* to ensure their controllable transition into a functional amyloid state and formation of long-term memory, and possibly a similar mechanism exists in other animals. This effect likely relates to the fact that many human amyloids are formed by proteolytic peptides rather than full-sized proteins.

Interesting and unexpected structural transitions occurred when various sequences were placed at the Sup35 N-terminus ahead of its PFR. Firstly, these sequences had a clear preference over the Sup35 PFR despite being seeded with native Sup35 prion. Even poorly amyloidogenic random extensions could form amyloid structure, though not in all cases. This could only be attributed to the advantage of the terminal location. Secondly, the formation of amyloid structure in the N-terminal extensions in all cases strongly altered and reduced the prion structure within the Sup35 N-terminal PFR, the structure that was used to seed the Sup35 extended versions. Thus, the N-terminal location of a sequence allows it to dominate in defining the prion fold. However, it is not clear why this should reduce the prion structure at its original location. This question is left for future studies.

## Figures and Tables

**Figure 2 ijms-26-01637-f002:**
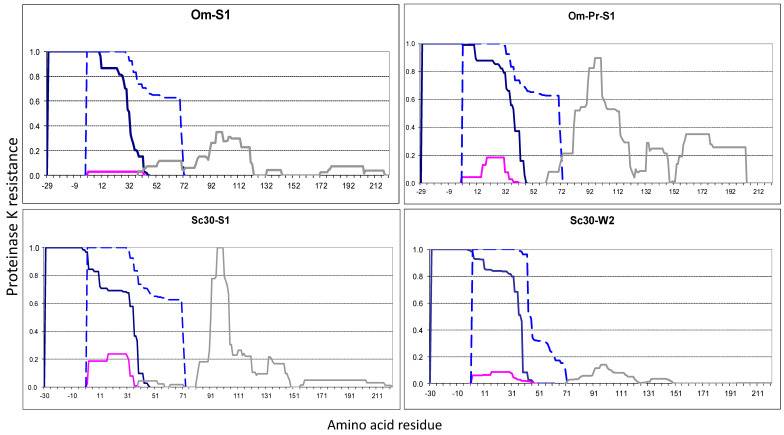
PK-resistant structures of Om-Sup35 and Sc30-Sup35 in the indicated [*PSI*+] strains. Om-Prion/S1: the S1 strain with chromosomal replacement of the *SUP35* gene for *Om*-*SUP35*. The graphs were obtained by summation of the amounts (peak areas) of the identified PK-resistant peptides according to their coordinates as in [10]. The obtained PK resistance values (Y-axis) were normalized, such that the maximal value would equal 1. The X-axis shows residue numbering according to the Sc Sup35 sequence, while the extensions have negative numbering. The major N-terminal Om-Sup35 structures are given in dark blue. The structure of the native Sup35 in [*PSI*+]-S1 strain is shown by the dashed blue line. The minor structures of uncertain origin (see text) are in magenta, and the grey line shows inner structures that were not considered in this work.

**Figure 3 ijms-26-01637-f003:**
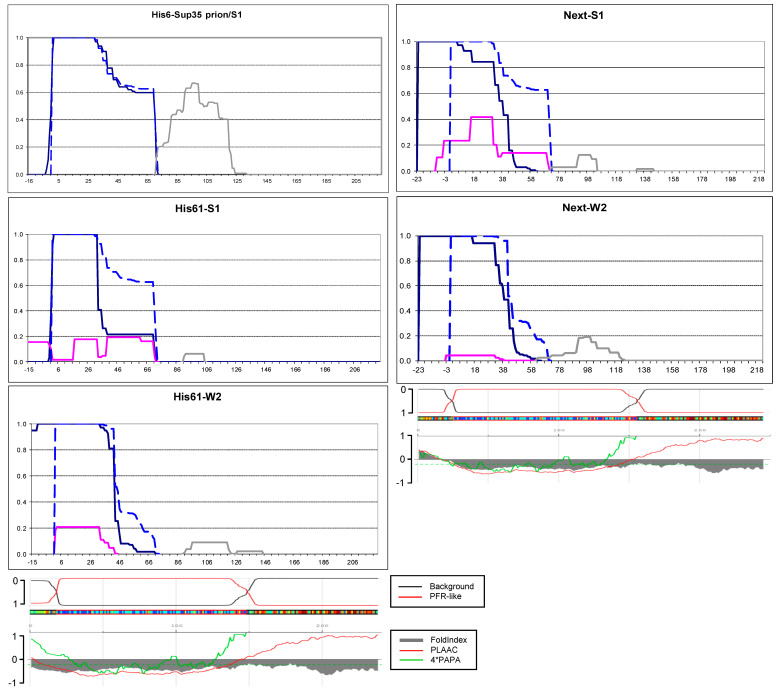
PK-resistant structures of His6-Sup35 and Next-Sup35 in the indicated [*PSI*+] strains. His6-Prion/S1: the S1 strain with chromosomal replacement of the *SUP35* gene for *His6*-*SUP35*. The graphs were obtained and annotated as in Figure 2. Computer predictions of the sequence propensity to form prion and non-prion structure were taken from https://plaac.wi.mit.edu/ (accessed on 29 November 2024) and aligned with the graphs.

**Figure 4 ijms-26-01637-f004:**
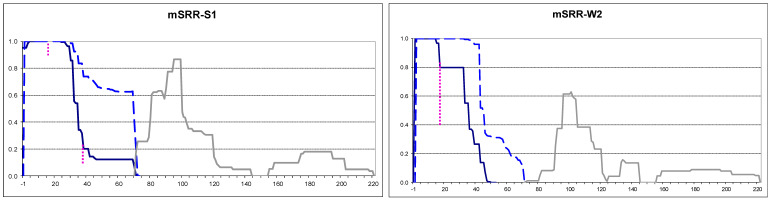
PK-resistant structures of mSRR-Sup35 indicated in the S1 and W2 [*PSI*+] strains. The graphs were obtained and annotated as in Figure 2. The extent of point cuts by PK appearing after residues 16 and 38 is indicated by dotted lines. When two peptides are adjacent with no gap between them, this gap is not visible in our Excel-based data processing procedure.

**Figure 5 ijms-26-01637-f005:**
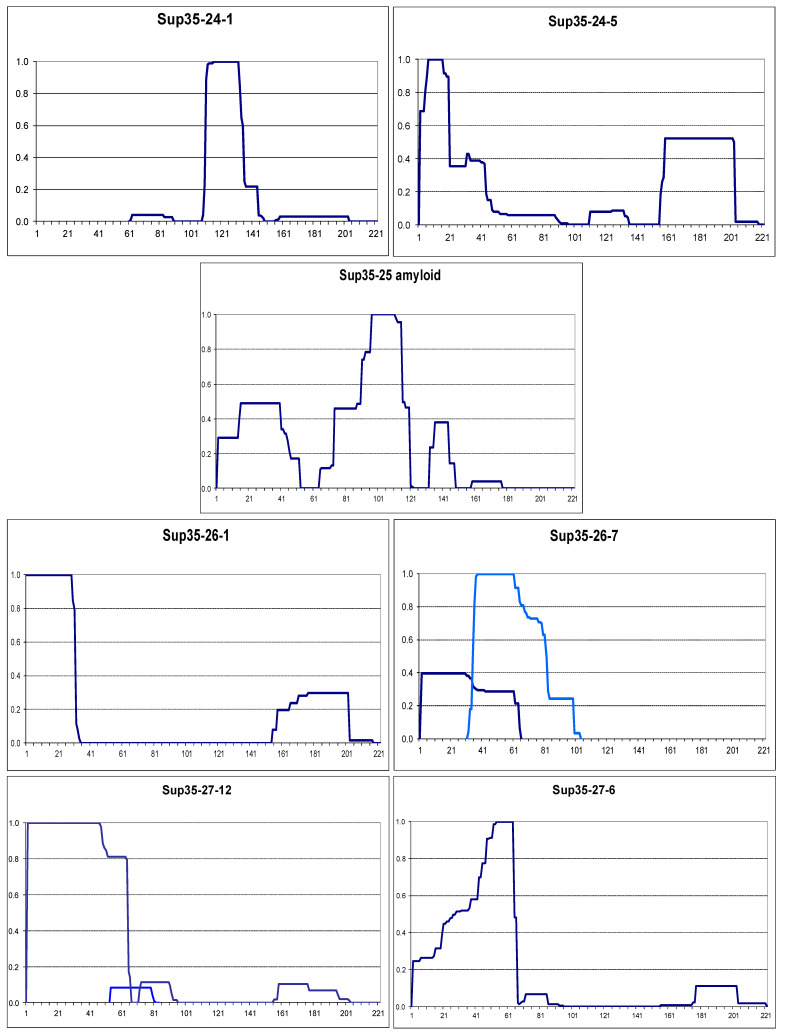
PK-resistant structures of prion forms of scrambled Sup35 proteins. The graphs were obtained and annotated as in Figure 2. Simultaneously, existing alternative structures are shown in different colors.

**Figure 6 ijms-26-01637-f006:**
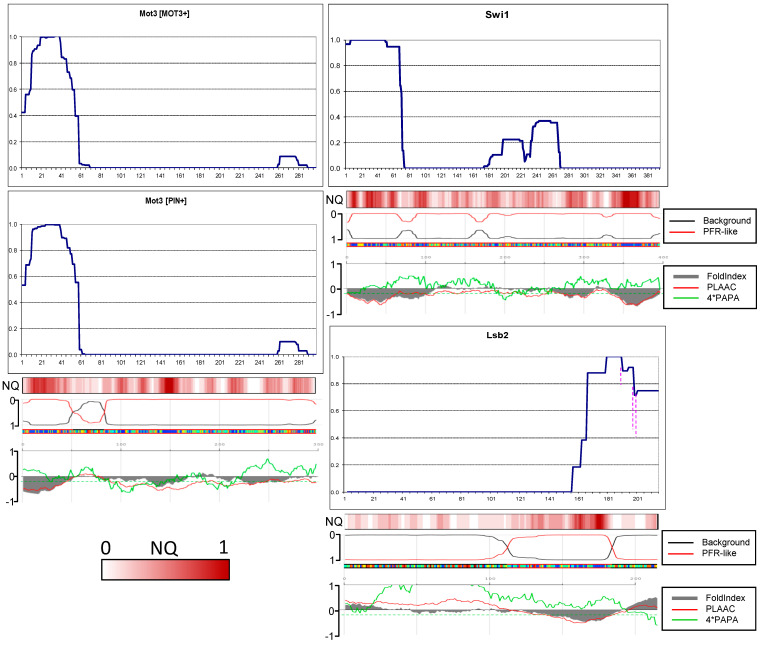
Prion and amyloid structures of the yeast prionogenic proteins. PK resistance maps were obtained and annotated as in Figure 2. The panels at the bottom are aligned to the maps and show the NQ content and prediction of prionogenicity and folded structure generated at http://plaac.wi.mit.edu/ (accessed on 29 November 2024).

**Figure 7 ijms-26-01637-f007:**
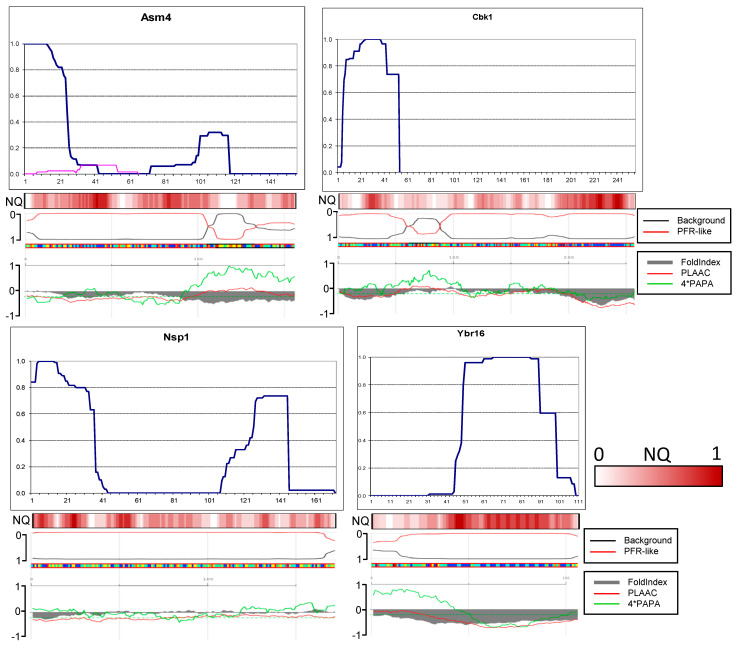
Amyloid structures of the yeast candidate prion proteins. PK resistance maps were obtained and annotated as in Figure 2. The panels at the bottom are aligned to the maps and show the NQ content and prediction of prionogenicity and folded structure generated at http://plaac.wi.mit.edu/ (accessed on 29 November 2024).

**Figure 8 ijms-26-01637-f008:**
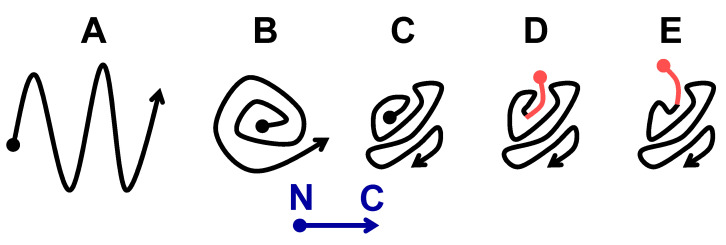
Possible prion core folding variants of Sup35 protomers. The N-terminus is indicated by a dot, and C-terminus by an arrowhead. (**A**) “Serpentine” folding according to [38]. (**B**) “Roll”. (**C**) Folding with a hidden N-terminus proposed in this work. (**D**,**E**) MSPPP-Sup35 structure (the extension is shown in pink) when seeded by the [*PSI*+] prions (**D**) or appearing de novo in the presence of [*PIN*+] (**E**).

**Figure 9 ijms-26-01637-f009:**
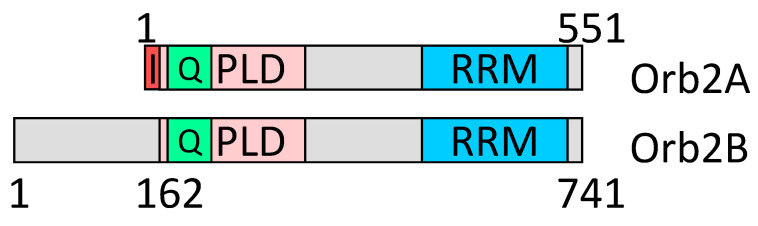
The Orb2 proteins maps. Q: the Q-rich region for which the amyloid structure is established with atomic resolution [40]. PLD: prion-like domain; RRM: RNA binding. I: eight-residue N-terminal sequence facilitating the Orb2A transition to amyloid form.

**Table 1 ijms-26-01637-t001:** PK resistance of the N-terminal residues of MSPPP-Sup35 when produced in the [*PSI*+]-S1, -W2, and [*PIN*+] background.

Residue	#	S1	W2	[*PIN*+]
M	−4	0	0	0
S	−3	0.94	0.96	0.29
P	−2	1	1	0.66
P	−1	1	1	0.94
P	1	1	1	1
S	2	1	1	1
D	3	1	1	1
S	4	1	1	1

The PK resistance was calculated as in Figure 2.

## Data Availability

Data is contained within the article and Appendix A.

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
