# Peer review of "On the Significance of the Terminal Location of Prion-Forming Regions of Yeast Proteins"

_ijms, 2025, doi:10.3390/ijms26041637_

Round 1

Reviewer 1 Report

Comments and Suggestions for Authors

The manuscript addresses the topic of prions, a scientific subject of great interest to the scientific community as it still contains many unanswered questions. I recommend accepting the manuscript pending revisions.

-The topic of prions is highly relevant to the scientific community. However, the authors included only one reference from 2023 and one from 2024 in the introduction. The authors should update the introduction by incorporating recent findings on the topic.

-It would be interesting for the authors to include a figure illustrating the amyloid structure mentioned in the introduction, highlighting the regions undergoing changes. The introduction currently consists solely of text.

-The authors did not associate experimental errors with the measurements. This should be added to the manuscript.

-The panels in Figures 6 and 7 are of very poor quality. The authors should improve their quality to enhance visualization.

-For the experimentally obtained structures, the authors could compare their results with predictions. There is a web server called BetaSerpentine, which could serve as a comparison tool.

-In the methodology section, the authors did not specify whether duplicates or triplicates were performed. This information should be included in the manuscript.

-In the discussion, the authors did not compare their results with data from the literature. They merely described their findings in an expository manner. The authors should compare their results with existing literature.

-The medium in which the protein is located influences the formation and stability of amyloids due to molecular interactions. The authors should discuss the role of these molecular interactions (e.g., 10.3390/molecules28196891 and 10.3390/ijms252312664).

-The authors did not provide a conclusion section for their results. A conclusion section should be included in the manuscript.

Author Response

Authors: We are grateful to this Reviewer for thorough review of our manuscript that helped to improve it. We have answered all the concerns and hope that the answers are satisfactory.

The manuscript addresses the topic of prions, a scientific subject of great interest to the scientific community as it still contains many unanswered questions. I recommend accepting the manuscript pending revisions.

Comment 1: -The topic of prions is highly relevant to the scientific community. However, the authors included only one reference from 2023 and one from 2024 in the introduction. The authors should update the introduction by incorporating recent findings on the topic.

Authors reply: We have added several recent citations to the introduction including one very good work that came out while we were making the revisions (PMID: 39656207).

Comment 2:-It would be interesting for the authors to include a figure illustrating the amyloid structure mentioned in the introduction, highlighting the regions undergoing changes. The introduction currently consists solely of text.

Authors: We have added the Sup35 map with amyloid structures to the Figure 1.

Comment 3:-The authors did not associate experimental errors with the measurements. This should be added to the manuscript.

-In the methodology section, the authors did not specify whether duplicates or triplicates were performed. This information should be included in the manuscript.

Authors: The only value calculated in this work is the Next-Sup35 level. It was calculated basing on five independent lysates of the Next-Sup35 cells (Figure S1). We have added this information to the Methods section. 

Comment 4:-The panels in Figures 6 and 7 are of very poor quality. The authors should improve their quality to enhance visualization.

Authors: The Figures in the best quality were provided separately as a Powerpoint file. The Figures within the text are for reviewing purposes only. We are big fans of the IJMS/MDPI format where the figures are given within the text. However, moving figures from Powerpoint to Word often results in distortions and other complications. Attempts to insert Figures 6 & 7 caused Word to crash. So, we inserted these figures as png images. We are forced to leave the task of perfect insertion of the Figures to technical Editors. Also we cannot agree about the "very poor quality". When printing these Figures to A4 sheet, it is difficult to notice any imperfections.

Comment 5: -For the experimentally obtained structures, the authors could compare their results with predictions. There is a web server called BetaSerpentine, which could serve as a comparison tool.

Authors: On one hand, such comparison was already provided where it was most required, i.e. in Figures 3, 6, 7. On the other, we tried to follow your advice and tried the BetaSerpentine. With it, we encountered two problems. Firstly, the offered output is not suitable for publication, since numerous structures are offered instead of what is usable for us: a function relating amino acid position and the probability of a prion structure. Secondly, we tested this algorithm on the Sup35 prion and the results were far from what we know from our PK resistance maps. However, at the same server at CRBM-CNRS Montpellier there are many other algorithms for the same task. One of them was recommended to us by its developer, Dr. Kajava, the ArchCandy. This algorithm proved to be more suitable, and we present these additional comparisons as a new supplementary Figure S7. The predictions were partly good, partly poor, and many of them would be better if accounting for the terminal location as a factor increasing the probability of а prion structure.

Comment 6: -In the discussion, the authors did not compare their results with data from the literature. They merely described their findings in an expository manner. The authors should compare their results with existing literature.

Authors: In the discussion, we mentioned the functional amyloid Orb2 of Drosophila and now we have improved this fragment. We also link our findings to the fact that many human amyloids are formed by proteins fragments rather than full proteins. Some literature on yeast prions is also discussed.

Comment 7: -The medium in which the protein is located influences the formation and stability of amyloids due to molecular interactions. The authors should discuss the role of these molecular interactions (e.g., 10.3390/molecules28196891 and 10.3390/ijms252312664).

Authors: This is true, but we do not know what we can add in this respect. Just our speculations, which are not worth much. We familiarized ourselves with the works you suggest but we do not see how we can use them in the frame of our work.

Comment  8: -The authors did not provide a conclusion section for their results. A conclusion section should be included in the manuscript.

Authors: The Conclusions sections is now provided.

Reviewer 2 Report

Comments and Suggestions for Authors

The manuscript investigated the role of terminal positioning in prion-forming regions of yeast proteins. They systematically investigated various factors affecting prion structure, including terminal extension, mutations, and scrambled sequences of Sup35. The authors showed that small changes near the N-terminus can disrupt prion propagation and larger extension (eq. 15-30 residues) often altered or replaced existing prion structure in Sup 35. They also investigated the amyloid structures of prion and candidate prion proteins. However, the authors found that the N-terminal residues of Swi1, Lsb2, Asmm4, and Nsp1 were fully protected, with minor reservations. Thus, the authors should be cautious to generalize that the terminal location of a sequence significantly increases its propensity to form amyloid, while international location has an opposite effect. The following are the suggestions that may improve the manuscript.

1.       The authors extended the discussion of Orb2 protein in Drosophila. Since there is no explanation for this protein, the authors should explain more about this protein. And the authors may add other proteins in different species, such as in human and other species to see whether they may reach a similar conclusion.

2.       The authors should explain the proteins Asm4, Swi1, Lab2, Nsp1, Mot3, Cbk1, Ybr16, etc. more in the manuscript. (Section 2.6). It is not easy to understand the content.

3.       Minor – Fig. 2 (Sc30-S1, check the X-axis label. No decimal point of amino acid residue number). For example, -10.0 should be -10.

Author Response

Authors: We are grateful to this Reviewer for thorough review of our manuscript that helped to improve it. We have answered all the concerns and hope that the answers are satisfactory.

Comment 1: The authors extended the discussion of Orb2 protein in Drosophila. Since there is no explanation for this protein, the authors should explain more about this protein. And the authors may add other proteins in different species, such as in human and other species to see whether they may reach a similar conclusion.

Authors reply: We have added to the discussion of Orb2 the description of its role and the way it functions, improved the text. Also added the consideration that many human amyloids are composed of proteolytic peptides, rather than full proteins, which shows that the same peptides inside complete proteins are reluctant to form amyloid.

Comment 2: The authors should explain the proteins Asm4, Swi1, Lsb2, Nsp1, Mot3, Cbk1, Ybr16, etc. more in the manuscript. (Section 2.6). It is not easy to understand the content. Also, Ybr16 or Ybr016w was renamed to Cpp1 according to recent suggestions in literature.

Authors: We have added a paragraph describing these proteins to the Section 2.6.

Comment 3: Minor – Fig. 2 (Sc30-S1, check the X-axis label. No decimal point of amino acid residue number). For example, -10.0 should be -10.

Authors: This was a small technical bug with Excel, which we could not fix quickly. Now this is corrected.

Round 2

Reviewer 2 Report

Comments and Suggestions for Authors

I want to recommend this manuscript to be published in the IJMS. The revised manuscript was satisfied to the reviewer.